# Intralesional steroid injections to prevent refractory strictures in patients with oesophageal atresia: study protocol for an international, multicentre randomised controlled trial (STEPS-EA trial)

Chantal A ten Kate ,[1] John Vlot,[1] Hanneke IJsselstijn,[1] Karel Allegaert,[2] Manon C W Spaander,[3] Marten J Poley,[4] Joost van Rosmalen,[5] Erica L T van den Akker,[6] Rene M H Wijnen[1]

For numbered affiliations see end of article.

**Correspondence to**
Dr John Vlot;
john.vlot@erasmusmc.nl

## ABSTRACT

**Introduction** Anastomotic stricture formation is the most common postoperative complication after oesophageal atresia (OA) repair. The standard of care is endoscopic dilatation. A possible adjuvant treatment is intralesional steroid injection, which is thought to inhibit scar tissue formation and thereby to prevent stricture recurrence. We hypothesise that this intervention could prevent refractory strictures and reduce the total number of dilatations needed in these children.

**Methods and analysis** This is an international multicentre randomised controlled trial. Children with OA type C (n=110) will be randomised into intralesional steroid injection followed by balloon dilatation or dilatation only. Randomisation and intervention will take place when a third dilatation is performed. The indication for dilatation will be confirmed with an oesophagram. One radiologist—blinded for randomisation—will review all oesophagrams. The primary outcome parameter is the total number of dilatations needed with <28 days' interval, which will be analysed with a linear-by-linear $\chi^2$ association test. Secondary outcome parameters include the level of dysphagia, the luminal oesophageal diameter and stricture length (measured on the oesophagrams), the influence of comedication on stricture formation, systemic effects of intralesional steroids (cortisol levels, length and weight) and the cost-effectiveness. Patients will undergo a second oesophagram; length and weight will be measured repeatedly; a scalp hair sample will be collected; and three questionnaires will be administered. The follow-up period will be 6 months, with evaluation at 2–3 weeks, 3 and 6 months after the intervention.

**Ethics and dissemination** Patients will be included after written parental informed consent. The risks and burden associated with this trial are minimal. The institutional review board of the Erasmus Medical Centre approved this protocol (MEC-2018–1586/NL65364.078.18). The results of the trial will be published in a peer-reviewed scientific journal and will be presented at international conferences.

## Strengths and limitations of this study

► This is the first prospective study investigating the effectiveness of intralesional steroid injection combined with endoscopic dilatation for the treatment of recurrent oesophageal strictures in children operated on for oesophageal atresia (OA).

► The determination of long-term cortisol levels over a period of 3 months will give objective information on possible systemic effects of intralesional steroid injection in children with OA.

► Besides the effectiveness and safety of intralesional steroid injection, cost-effectiveness of this treatment will be evaluated.

► The rarity of the disease makes patient recruitment challenging.

**Trial registration numbers** 2018-002863-24 and NTR7726/NL7484.

## INTRODUCTION

Oesophageal atresia (OA) is a congenital malformation which can present with or without a tracheoesophageal fistula, with a European prevalence of 2.43 cases per 10 000 births.[1 2] With better treatments, survival rates have increased to over 90%.[1 3 4] Still, anastomotic stricture formation remains the most frequent postoperative complication in up to 60% of cases.[5] Especially refractory strictures form a great burden for both patients and their parents.

The incidence of refractory strictures is poorly reported due to the variety in definitions used in literature. Regarded as a consensus among experts, the European Society for Paediatric Gastroenterology

Hepatology and Nutrition (ESPGHAN) has set the following definition for a refractory oesophageal stricture: an anatomical restriction without endoscopic inflammation that results in dysphagia after a minimum of five dilatation procedures at maximally 4-week intervals.[6 7]

Recently, the Dutch Consortium for Esophageal Atresia (DCEA) conducted a retrospective multicentre study in the Netherlands to assess risk factors for stricture formation in children with OA. The study population consisted of 436 children born with OA between 1999 and 2013 with an end-to-end oesophageal anastomosis. Thirty-two (7.3%) of them required ≥5 dilatations within an interval of 28 days.[5]

The initial treatment of an anastomotic stricture consists of endoscopic dilatation, either balloon dilatation or semirigid dilatation.[6] Consensus on the preferred technique has not yet been established. A refractory stricture requires multiple dilatations under general anaesthesia, for which the child needs to be hospitalised. This adds significantly to the burden of the disease. It is therefore important to minimise the occurrence of refractory strictures and with that the need for dilatations.

In a recent ESPGHAN guideline, various adjuvant treatments are mentioned, for example, intralesional or systemic steroids, topical mitomycin C, oesophageal stents and surgical resection. Our trial will focus on intralesional steroid injections since we as well as the other centres involved in this trial have had successful results with this treatment in several patients.[8]

The literature on intralesional steroid injection in children with OA is scarce, but promising results have been described in both children and adults with all types of oesophageal strictures (table 1). Most studies are outdated case reports or series.[9–14] Four relatively recent randomised controlled trials (RCTs) on this topic included only adults with underlying diagnoses other than OA, like caustic strictures after acid ingestion, peptic strictures and anastomotic strictures after oesophagectomy with gastric tube reconstruction.[15–18] Reported beneficial effects were reduction of dilatation procedures,[16] longer intervals between dilatation procedures,[17] improvement of luminal diameter[15] and relief of dysphagia.[18]

In children, only retrospective studies have been performed, which mostly included caustic strictures.[19 20] Divarci et al analysed data of 32 children with corrosive strictures with a mean age of 3.6 years (±2.5 years); after the intervention, the number of dilatations had significantly decreased and the intervals between dilatations had extended.[19] Cakmak et al included 38 children with either OA or corrosive strictures with a median age of 1.5 years (range 0–14 years)[20] but did not find a significant difference in treatment effectiveness. None of the above-mentioned studies reported any systemic effects of the local intralesional steroid injections.

All studies used triamcinolone acetonide (TAC). The exact mechanism by which TAC enhances the efficacy of dilatation is unclear. It has been proven very effective in the treatment of hypertrophic scars of the skin and keloid. A recurring anastomotic stricture can be seen as a hypertrophic lesion. The injected TAC inhibits collagen formation, enhances collagen breakdown, decreases fibrotic healing that occurs after dilatation and prevents cross-linking of collagen that causes contractions in scar tissue.[21 22]

The recurrent dilatations and readmissions impose a substantial burden on the healthcare system. To date, there is no evidence on the cost-effectiveness of intralesional steroid injections. However, in the current era of evidence-based and cost-effective medicine, proof of cost-effectiveness is highly relevant.

The primary objective was to evaluate whether intralesional steroid injections combined with endoscopic dilatation can prevent refractory strictures in children with OA and recurrent oesophageal stenosis, and thus can minimise the number of dilatations needed with a 28 days' interval between the dilatations.

The secondary objectives were
► To compare the level of dysphagia and the child's eating behaviour between the two groups.
► To compare the effect of intralesional steroid injections on the luminal diameter and the stricture length between the two groups.
► To evaluate a possible influence of comedication (eg, antacids) on stricture formation.
► To analyse the possible systemic effects of a one-time intralesional steroid injection.
► To analyse the cost-effectiveness of the use of intralesional steroid injections to prevent refractory strictures.

## METHODS AND ANALYSIS
The STEPS-EA trial is an international, multicentre, single-blinded RCT with a 1:1 randomisation to injection with 10 mg/mL TAC (Kenacort-A 10) prior to balloon dilatation and balloon dilatation without any injection. The participating centres are tertiary (academic) hospitals that collaborate within the European Reference Network on Inherited and Congenital Abnormalities (ERNICA, ern-ernica.eu) and that routinely provide care for children with OA.

### Patient and public involvement
Parents of patients were involved in the end stage of the design of the trial. We presented our plans for the trial to a DCEA meeting in which also representatives of the patients' association Vereniging voor Ouderen en Kinderen met een Slokdarmafsluiting (VOKS) took part. They were invited to comment on the study design, intervention or time required to participate in this trial. Consensus was reached on the final design during this meeting. The patients' association will not be involved in the recruitment and conduct of the trial. We will involve them in dissemination, however, by presenting the trial results at a members' day or in their monthly newsletter. As the VOKS is a member of the European Federation of Esophageal Atresia and Tracheoesophageal fistula

ten Kate CA, et al. BMJ Open 2019;9:e033030. doi:10.1136/bmjopen-2019-033030

**Table 1** Summary of literature on clinical findings on intralesional steroid injections for oesophageal strictures, including retrospective cohort studies in children <12 years (range 0–14 years)

| Author (year) | Type of study | Characteristics | Main outcomes |
|---|---|---|---|
| Camargo et al (2003)[15] | Double-blind RCT | 14 adult patients, corrosive strictures | ► No significant difference in dilatation frequency and dysphagia.<br>► Significant improvement in obtained diameter (p<0.05).<br>► No adverse events reported. |
| Ramage et al (2005)[16] | Double-blind RCT | 30 adult patients, peptic strictures | ► Less patients required repeat dilatation in the steroid group (13% vs 60%, p=0.0209).<br>► Shorter time to repeat dilatation in the control group (p=0.01).<br>► No adverse events reported. |
| Hirdes et al (2013)[17] | Double-blind RCT | 60 adult patients, anastomotic strictures after oesophagectomy with gastric tube reconstruction | ► No significant decrease in frequency of repeat dilatation or prolongation of dysphagia-free period.<br>► Four patients developed Candida oesophagitis. |
| Pereira-Lima et al (2015)[18] | Double-blind RCT | 19 adult patients, anastomotic strictures after oesophagectomy with gastric tube reconstruction | ► Significant improvement on dysphagia at 1 and 6 months (p=0.021 and p=0.009).<br>► No perforation or haemorrhage of oesophageal candidiasis, no other adverse events reported. |
| Kochhar and Makharia (2002)[35] | Prospective | 71 patients (13–78 year), all kinds of strictures | ► Periodic dilatation index decreased significantly after injection (p<0.001).<br>► No adverse events reported. |
| Nijhawan et al (2016)[36] | Prospective | 11 adult patients, corrosive strictures | ► Significant improvement of maximum dilatation (p<0.001) and number of dilatations per month (p<0.001).<br>► No adverse events reported. |
| Divarci et al (2016)[19] | Retrospective | 32 children (mean age 3.6 years), corrosive strictures | ► Mean number of dilatation sessions was decreased (p=0.003).<br>► Mean frequency of dilatations in weeks extended (p<0.001).<br>► Only a positive effect in short-segment strictures (<3 cm, 92% of patients dysphagia-free).<br>► No serious adverse events reported, one transient cushingoid phenotype but no real adrenal suppression. |
| Cakmak et al (2016)[20] | Retrospective | 38 children (median age 1.5 years), OA (n=19) and corrosive strictures (n=19) | ► No significant difference in treatment effectiveness between steroid injection and others (p>0.05).<br>► Intralesional steroid injections performed only in patients with long (>5 cm) and corrosive strictures and ≥5 dilatations.<br>► Four patients with oesophageal perforation at other dilatation sessions than the intralesional steroid injection. |

OA, oesophageal atresia; RCT, randomised controlled trial.

Support Groups E.V., patients and parents throughout Europe will be informed about the results.

### Participants

OA can be a very heterogeneous disease. About 90% of the children with OA has type C.[5 23] In order to make the two treatment groups as equal as possible, only children with OA type C who underwent surgery with primary anastomosis within the first days of life and who developed a recurrent oesophageal stricture will be eligible. Children will be included if they are ≥3 months old at the time of the intervention and in need of a third dilatation. Written parental informed consent will be obtained by the local principal investigator (PI) or another member of the local research team. Exclusion criteria are lack of parental consent or an impossibility—known from previous dilatations—to use an endoscope

**Table 2** Assumed relative frequencies of the number of dilatations in the control and steroid groups

| Number of dilatations within 28 days' interval | Observed number of patients (n=407)[5] | Relative frequencies, control group | Assumed relative frequencies, steroid group |
|---|---|---|---|
| 3 dilatations | 4 | 0.075 | 0.142 |
| 4 dilatations | 7 | 0.132 | 0.302 |
| 5 dilatations | 9 | 0.170 | 0.170 |
| 6 dilatations | 7 | 0.132 | 0.160 |
| 7–10 dilatations | 16 | 0.302 | 0.132 |
| >10 dilatations | 10 | 0.189 | 0.094 |
| Total (all numbers of dilatations combined) | 53 | 1.000 | 1.000 |

with a large enough diameter working channel to pass the endoscopic injector.

### Sample size calculation

The power calculation is based on a linear-by-linear $\chi^2$ association test comparing the total number of dilatations required within the study period (all strictures) and within a 28-day interval (refractory strictures) between the two treatment groups. The total number of dilatations will be categorised into the categories 3, 4, 5, 6, 7–10 and >10 dilatations, and a separate category for patients who are not dysphagia-free at the end of the follow-up period.

For this power calculation, we used data of the original dataset of our retrospective study in the Netherlands.[5] We selected patients from this dataset who underwent at least three dilatations with a 28-day interval (n=53). The retrospective study's observed numbers of patients and the relative frequencies for each category are listed in table 2. We assumed that the use of intralesional steroid injections combined with endoscopic dilatation will reduce the total number of dilatations by 50%. Note that this 50% reduction applies only to dilatations after the third dilatation, and therefore no change in the number of dilatations is assumed for the first three dilatations. This assumption leads to a different distribution for the number of dilatations within the categories, which is shown as the assumed relative frequencies for the steroid group in the final column of table 2. The details of the calculation are provided in online supplementary file 1.

In a simulation model, the required sample size to obtain a power of 80% (with a two-sided significance level of 0.05) was calculated as 52 patients per group, thus 104 in total. To account for the effects of dropout and missing data, we aimed to include a total of 110 patients.

### Recruitment

Patients will be recruited from hospitals in various European countries. Up until now, hospitals in Denmark, UK, Finland, France, Italy and Sweden have agreed to participate. During the inclusion period, it will remain possible for other centres to join. To achieve adequate participant enrolment, we have minimalised the exclusion criteria. Collaboration within ERNICA should make patient accrual achievable despite the rarity of the disease.

### Randomisation, blinding and treatment allocation

Randomisation will be conducted via ALEA (FormsVision B.V./ALEA Clinica B.V.), a validated software program. To achieve equal distribution of the intervention among the participating sites, block randomisation stratified per centre will be carried out. The software was prepared by an independent statistician who is not otherwise involved in the study. After inclusion, the local PI will enter the patient in ALEA and will thereupon receive an email stating the allocated treatment.

Randomisation will be blinded for the radiologist who will review all oesophagrams. The control group will not receive sham treatment. The steroid that will be used, Kenacort-A 10 (see further), is a white suspension, which complicates creating a placebo. Adding excipients to normal saline is undesirable, considering the unknown effect on the healing process of the stricture. Moreover, it is deemed undesirable to inject an infant with a fluid with an unknown effect.

### Investigational product

The intervention will be a one-time endoscopic injection of 0.25 mL Kenacort-A 10 (Bristol-Myers Squibb BV, Utrecht, the Netherlands) in each quadrant of the stricture prior to the third endoscopic dilatation. Thus, in total, 1 mL (10 mg/mL TAC, equals 12.5 mg prednisone) will be injected. During the study period, none of the patients will receive a second injection.

Kenacort-A 10 will be prepared, labelled and distributed by the trial pharmacy of the coordinating hospital. It will be delivered to the local pharmacies by courier, and the local pharmacy will deliver it to the operation room when needed. After the trial, a specific procedure for destruction of the remaining drugs is not needed; they can be disposed of locally.

### Patient timeline

Figure 1 presents a flowchart for this study and the study procedures. The treating physician will decide on a third dilatation on the basis of the clinical signs of dysphagia and the findings on the oesophagram (thoracic X-ray with contrast, anterior–posterior and lateral). Clinical signs of dysphagia are defined as the inability to be fed age appropriately. Findings on the oesophagram indicating

ten Kate CA, *et al. BMJ Open* 2019;**9**:e033030. doi:10.1136/bmjopen-2019-033030

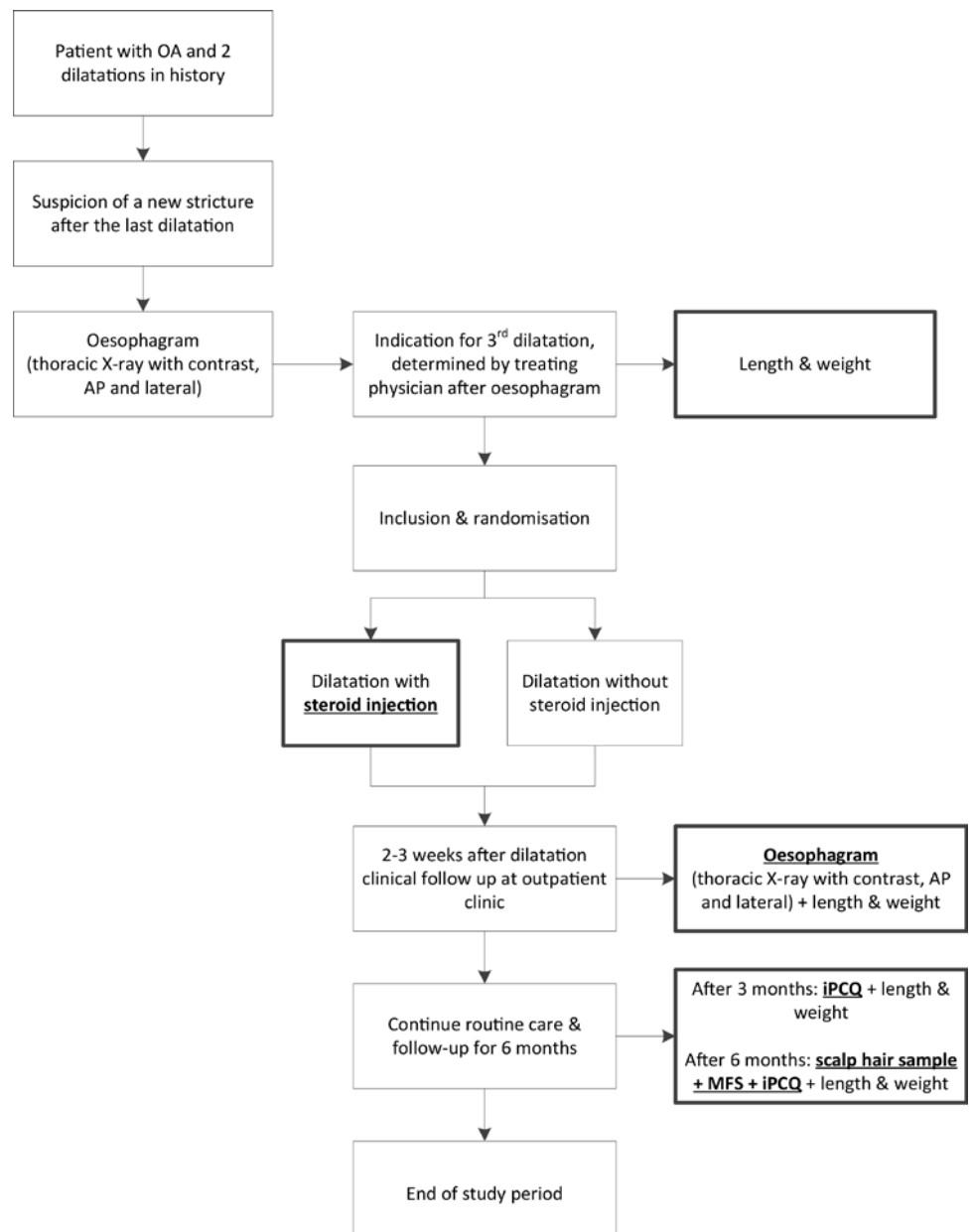

**Figure 1** Flowchart of the study design. Bold with underline indicates study procedures; the rest is standard of care. AP, anterior–posterior; iPCQ, iMTA Productivity Cost Questionnaire; MFS, Montreal Feeding Scale; OA, oesophageal atresia.

a stricture are defined as a significant narrowing of the lumen, seen as a waist in the contrast on the X-ray. The treating physician can be a paediatric surgeon or a paediatric gastroenterologist, depending on the local agreements in the different countries. After parental informed consent, the patient will be included and randomised to one of the study arms.

Prior to the balloon dilatation, an endoscopic needle (DVI-23-MH varices injector or equivalent) is prefilled with 1 mL Kenacort-A 10 and passed through the endoscope. Under direct vision, 0.25 mL will be injected in each of the four quadrants of the circular stricture. After the injection needle has been retracted from the working channel and good visualisation has been re-established,

balloon dilatation will be performed up to the desired diameter. The balloon will remain insufflated for 1 min.

During a follow-up period of 6 months, patients will not receive any (additional) steroid injections, only balloon dilatations if needed. Meaning, patients in the steroid group will only receive one injection, and patients in the control group will receive no injections at all. After the study period has ended, treatment is again free of choice.

Two to 3 weeks after the dilatation procedure, a second oesophagram will be made in children in both study groups. All oesophagrams will be reviewed by one specialised paediatric radiologist of the coordinating hospital, who will determine the oesophageal diameter and stricture length. Earlier studies have proven that these

measurements can be obtained from an oesophagram.[24] The parents will be informed about the results and the normal standard of care will be continued.

Each patient will be followed up for 6 months after the third dilatation, with recording of possible side effects, complications and additional dilatations. Length and weight will be measured at 2–3 weeks, 3 and 6 months after the third dilatation. In the context of the cost-effectiveness analysis, the parents will be asked to fill out a modified version of the iMTA Productivity Cost Questionnaire (iPCQ)[25] at 3 months after the third dilatation.

At evaluation at 6 months, we will collect a scalp hair sample from the child to determine long-term cortisone and cortisol levels.[26 27] The hair locks will be stored for final batch analysis. Lastly, the parents will be asked to fill out the Montreal Feeding Scale (MFS)[28–30] and again the iPCQ. After this, the study period will end.

### Outcome parameters

The primary outcome parameter is the total number of dilatations required per patient with a 28-day interval between the dilatations during the study period of 6 months, which is defined as the period from the day of the third dilatation until 6 months later.

The secondary outcome parameters are as follows:

1. Total number of dilatations within the study period, regardless of the interval.
2. Interval (in weeks) between the start of the study and the last dilatation procedure within the study period.
3. Scores on the MFS.
4. The change in maximal luminal diameter after the third dilatation relative to the diameter before the third dilatation: relative change in luminal diameter=(maximal diameter after–maximal diameter before)/maximal diameter before. The diameter will be measured at the narrowest point of the oesophagus.
5. The change in the length of the oesophageal stricture after the third dilatation relative to the length before the third dilatation: relative change in stricture length=(stricture length after–stricture length before)/stricture length before. The length will be measured between the two points where the oesophageal diameter starts narrowing.
6. The use of comedication (eg, antacids) during the study period.
7. The mean hair cortisol levels in the first 3 months after the third dilatation. Cortisol levels will be adjusted for age and sex.
8. Delta length SD scores (SDSs) and delta weight SDS between the third dilatation (intervention) and 3 and 6 months after the third dilatation.
9. Total costs of the treatment, including medical and non-medical costs.
10. Incremental costs per refractory stricture prevented and incremental costs per additional dysphagia-free patient.

### Data collection

All participating centres are familiar with the procedure of injecting Kenacort-A 10 in the lesion via an injection needle through the endoscope. However, to guarantee equality of the intervention, the relevant practitioners in all centres will be trained by the PI of the trial. Radiological interobserver variability will be avoided by having all oesophagrams reviewed by one radiologist.

The internationally validated MFS[28–30] will be used to measure dysphagia. The MFS has often been used in previous research in children with OA. The scalp hair sample will be taken from the posterior vertex; cortisol levels will be determined with the liquid chromatography–tandem mass spectrometry method for quantification of steroids.[26]

The cost-effectiveness analysis will follow established methods for economic evaluations and costing studies in healthcare.[31 32] Both medical and non-medical costs will be collected. Medical costs will include costs of surgeries (dilatations), steroid injections, hospital days (on the ward or the intensive care unit), medication (such as analgesics) and diagnostic radiography. Costs of healthcare provided by others than the participating centres (such as other hospitals or general practitioners) will be ignored in this study, as these are unlikely to be affected by the intervention. Non-medical costs will include costs of special diets, costs related to hospital visits and productivity losses related to both paid and unpaid works. The non-medical costs will be measured using the iPCQ,[25] supplemented with additional questions on costs of special diets and costs related to the child's hospitalisation. The original iPCQ questions are validated in English. The complete questionnaire including the additional questions will be translated to the languages required for this trial using the forward–backward translation method, as will the MFS for the languages it has not been validated for.

### Statistical analysis

Since a standardised treatment protocol will be adopted in all centres, statistical adjustment for centre effects is considered unnecessary.

The primary outcome parameter will be analysed with a linear-by-linear $\chi^2$ association test. The total number of dilatations required with a 28-day interval (ie, refractory strictures) within the study period of 6 months per patient will be categorised and compared between treatment groups. In case of death during the follow-up period, the outcome will be set to the highest (ie, most severe) category. In case of drop-out during the follow-up period due to other causes (eg, emigration and withdrawal), the subject will be excluded from the study.

The analyses for the secondary study parameters are as follows:

1. The total number of dilatations required within the study period (regardless of the interval, ie, all strictures) will be categorised and compared between treatment groups with a linear-by-linear $\chi^2$ association test.

ten Kate CA, *et al. BMJ Open* 2019;**9**:e033030. doi:10.1136/bmjopen-2019-033030

2. The interval (in weeks) until the patient is dysphagia-free will be compared between groups with the log-rank test and with Cox proportional hazards regression, with adjustment for treatment group and factors such as age, sex, diagnostic information and described risk factors for stricture formation like anastomotic leaking and thoracoscopic repair.[5] Patients who do not become dysphagia-free during follow-up are treated as censored at the end of the follow-up period.

3. The scores on the MFS (reflecting the level of dysphagia and the eating behaviour) will be compared between study groups with a Mann-Whitney test.

4. The relative change in the oesophageal diameter and that in the length of the oesophageal stricture will be compared between study groups with an analysis of covariance model. The dependent variables in this model will be the log-transformed oesophageal diameter and length after the third dilatation, and the independent variables will be the treatment group and the log-transformed oesophageal diameter and length before third dilatation.

5. The effect of comedication on the primary study outcome will be assessed using a stratified Mann-Whitney test with stratification for the treatment group.

6. The mean cortisol level over the first 3 months after the third dilatation will be compared between study groups with a linear regression model with adjustment for age and sex.

No missing data are expected for the independent variables in the analysis of covariance models and the Cox proportional hazards regression models. In case of missing data for any of the outcomes, a complete case analysis (ie, exclusion of the subjects who dropped out during the follow-up period) will be performed for the corresponding outcome.

### Cost-effectiveness analysis

All medical and non-medical costs will be summated for each individual patient. Regarding the patient outcomes, the number of refractory strictures prevented and the number of dysphagia-free patients will be considered. Incremental cost-effectiveness ratios will be calculated, expressed as incremental costs per refractory stricture prevented and incremental costs per additional dysphagia-free patient. Analysis of uncertainty will be illustrated through cost-effectiveness planes (via bootstrapping).

As this RCT is done in an international setting, it must be recognised that the results may differ between countries because healthcare systems, treatment patterns and prices may vary. Therefore, country-level information will be collected. Data (especially resource quantities and cost prices) will be collected in all countries, and results will also be reported for each country separately. Next, a pooled summary calculation of the intervention's cost-effectiveness will be made, converting all costs into a common currency base (ie, euros).

### Adverse events and auditing

Adverse events will be handled according to the guidelines of the institutional review board (IRB) of the Erasmus Medical Centre. All adverse events will be registered during the study. Serious adverse events will be reported to the sponsor immediately and registered appropriately within 24 hours.

All participating sites will be audited once a year with monitoring of patient recruitment, source data verification, drug accountability and sample storage. The auditor will be independent and not involved in the study.

### Benefits and risk assessment

The risks and burden associated with this study are minimal. Potential complications of oesophageal steroid injections include adrenal suppression, perforation, intramural infection, *Candida* infection, mediastinitis and pleural effusion.[33] However, in previous studies, no adverse events have been reported in relation to the steroid injections (see table 1). Additionally, Kenacort-A 10 is a slow-release medicine and therefore a gradual exposure. This implies that the likelihood of acute exposure to a high dose of steroids will be minimal.

The burden of filling out the questionnaires and taking a hair sample are negligible. Filling out the MFS and the iPCQ will take maximally 30 min. All participants will undergo one extra oesophagram after the third dilatation procedure. The potential reduction in the number of anaesthetic procedures needed for dilatations outweighs the burden and the radiation exposure of this oesophagram. Potential benefits of intralesional steroid injections are fewer dilatation procedures needed, with concomitant fewer anaesthetic procedures and hospital admissions and less risk of perforation.

### Data management

All data will be handled confidentially and anonymously using OpenClinica V.3.12.2 (OpenClinica LLC, USA) for data collection. The questionnaires will be conducted through an online survey using LimeSurvey (LimeSurvey GmbH, Germany) and GemsTracker (Erasmus MC, Rotterdam, the Netherlands). All patient data will be coded using a subject identification code list. The local PI safeguards the key to the code; the sponsor will have access to these codes. The local PI will only have access to the data of patients of their own centre; the sponsor will have access to the final trial dataset. All has been stated in a clinical trial site agreement signed by all participating sites.

### ETHICS AND DISSEMINATION

This study protocol was approved by the IRB of the Erasmus Medical Centre (MEC-2018–1586/NL65364.078.18). In case of any modifications of the protocol, a formal amendment will be submitted to the IRB. Approved changes will be communicated to all relevant parties according to the rules of the IRB. The protocol has currently

been submitted for local ethical approval in Amsterdam (the Netherlands), Odense (Denmark), Copenhagen (Denmark), Stockholm (Sweden), Rome (Italy) and Padua (Italy). Nijmegen (the Netherlands) and Helsinki (Finland) have already obtained local ethical approval and joined the study. To guarantee the respect of ethical rules and standard of care in all participating centres, the protocol was reviewed by the two chairs of the work package on OA within ERNICA. The informed consent and assent process of this trial is in line with the Good Clinical Practice guideline.[34]

Furthermore, we involved external experts in the form of a data safety monitoring board (DSMB). These experts are a paediatric surgeon, a pharmacologist and a statistician. All experts are independent of the sponsor and therefore competing interests will be avoided. The DSMB will monitor the safety of the study subjects and data. The board will meet at least three times: within 1 year of recruitment commencing, at the time of the planned interim analyses at 50% (n=55) of enrolment and at the conclusion of the trial.

The results of this trial will be published in an international peer-reviewed scientific journal, within 1 year after the end of the follow-up period of the last included patient. In addition, we aimed to present the results at several international conferences to inform healthcare professionals worldwide.

## TRIAL STATUS

The study has started in Rotterdam (the Netherlands) in February 2019 after ethical approval had been obtained. The first patient is included. Nijmegen (the Netherlands) has joined the study in October 2019 and Helsinki (Finland) in December 2019. At this moment, Amsterdam (the Netherlands), Odense (Denmark), Copenhagen (Denmark), Stockholm (Sweden), Rome (Italy) and Padua (Italy) are still waiting for local ethical approval. We expect to start the study in these sites the latest in 2020.

Considering the rarity of the disease, we expect to complete the inclusions and finish data collection for this study in 5 years.

**Author affiliations**
¹Department of Paediatric Surgery and Intensive Care, Erasmus University Medical Center, Sophia Children's Hospital, Rotterdam, The Netherlands
²Department of Paediatrics, Division of Neonatology, Erasmus University Medical Center, Sophia Children's Hospital, Rotterdam, The Netherlands
³Department of Gastroenterology and Hepatology, Erasmus University Medical Center, Rotterdam, The Netherlands
⁴Institute for Medical Technology Assessment, Erasmus University Rotterdam, Rotterdam, The Netherlands
⁵Department of Biostatistics, Erasmus University Medical Center, Rotterdam, The Netherlands
⁶Department of Paediatric Endocrinology, Erasmus University Medical Center, Sophia Children's Hospital, Rotterdam, The Netherlands

**Acknowledgements** The authors thank Benno Ure, paediatric surgeon and head of the Department of Paediatric Surgery of Hannover Medical School, and Frédéric Gottrand, paediatrician at the Department of Paediatric Gastroenterology of University Hospital of Lille, for their input during the development of the trial protocol; Joke Dunk, research nurse at the Erasmus University Medical Centre in Rotterdam, for her help with the implementation of the trial. Ko Hagoort provided editorial advice; and lastly, all hospitals involved in European Reference Network on Inherited and Congenital Abnormalities and the Vereniging voor Ouderen en Kinderen met een Slokdarmafsluiting for their input and collaboration.

**Contributors** RMHW is the initiator of the study. CAtK, JV, HIJ, KA, MCWS, MJP, JvR, ELTvdA and RMHW contributed in writing and designing the study protocol. JV and RMHW supervised the study. CAtK coordinated the study and was responsible for data acquisition. CAtK, JV and JvR carried out the analyses. All authors read and approved the final version of the study protocol and manuscript.

**Funding** The authors have not declared a specific grant for this research from any funding agency in the public, commercial or not-for-profit sectors.

**Competing interests** None declared.

**Patient consent for publication** Not required.

**Provenance and peer review** Not commissioned; externally peer reviewed.

**ORCID iD**
Chantal A ten Kate http://orcid.org/0000-0001-9921-7776

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
