## [Reviewer comments · BMJ Open]

ARTICLE DETAILS

TITLE (PROVISIONAL)	The STEPS-EA trial: intralesional STERoid injections to Prevent refractory Strictures in patients with Esophageal Atresia – study protocol for an international, multicentre randomised controlled trial
AUTHORS	ten Kate, Chantal; Vlot, John; Ijsselstijn, Hanneke; Allegaert, Karel; Spaander, Manon; Poley, Marten; van Rosmalen, Joost; van den Akker, Erica; Wijnen, Rene

VERSION 1 - REVIEW

REVIEWER	Omid Madadi-Sanjani Center of Pediatric Surgery Hannover Medical School Germany
REVIEW RETURNED	15-Aug-2019

GENERAL COMMENTS	I was pleased to read this study protocol, investigating an important problem in the postoperative care of infants with esophageal atresia. The main concern is the heterogenous expression of the EA types, leaving a main bias in the comparison. However, this is an unique attempt to investigate the problem of postoperative strictures in a prospective study, as desperately suggested by multiple authors. The protocol is detailed, the outcome parameters are accurate. Therefore, I don't have any major/ minor concerns regarding the study design. My recommendation is to accept the manuscript.
--

REVIEWER	Noboru Hanaoka Osaka Red Cross Hospital, Osaka, Japan
REVIEW RETURNED	08-Oct-2019

GENERAL COMMENTS	Thank you for giving me a chance to review this article. This is a study protocol of randomized controlled trial to evaluate whether intralesional steroid injections combined with endoscopic dilation can prevent refractory strictures in children with
---

	oesophageal atresia. I hope steroid injection would make a contribution to the reduction of dilations. Major comments A recent randomized controlled trial found that adding steroid injections before dilation did not result in a significant reduction of repeat dilations in patients with anastomotic strictures after esophagectomy (Herdes M et al. Endoscopic corticosteroid injections do not reduce dysphagia after endoscopic dilation therapy in patients with benign esophagogastric anastomotic strictures. Clin Gastroenterol Hepatol. 2013; 11:795-801.) However, another randomized controlled trial found that targeted steroid injection into the laceration after dilation reduce the need for repeat dilation (Hanaoka N et al. Endoscopic Balloon Dilation Followed By Intralesional Steroid Injection for Anastomotic Strictures After Esophagectomy: A Randomized Controlled Trial. Am J Gastroenterol. 2018;113:1468-1474.) However, no studies have compared injection before with injection after dilatation and this would be remained to be discussed. Why the authors use steroid injections prior to the dilatation? Minor comments  1. According to the Patient timeline, the treating physician will decide on a third dilatation on the basis of the clinical signs of dysphagia and the findings on the esophagram. What are the clinical signs of dysphagia and findings on the esophagram? Are there any criterions for third and additional dilations such as dysphagia score? The authors should clarify the indication of third and additional dilatations precicely. Need details on your dysphagia scale and is it a validate score? 2. In method and analysis, the authors mentioned that the intervention is an endoscopic injection of Kenacort in each quadrant of the stricture prior to the third endoscopic dilatation. Howerver, the order of injection and dilation was reversed in abstract. According to the abstact, the patients will be randomized into balloon dilatation followed by intralesional stesoid injection or dilation only. 3. Were the multiple triamcinolone injections allowed - or just the initial treatment? I suspect initial, but need to clarify
--	---

VERSION 1 – AUTHOR RESPONSE

Reviewer 1:

I was pleased to read this study protocol, investigating an important problem in the postoperative care of infants with oesophageal atresia. The main concern is the heterogeneous expression of the EA types, leaving a main bias in the comparison.

First, we would like to thank you for the careful appraisal of our manuscript. We agree with the reviewer that oesophageal atresia can indeed be a very heterogeneous disease. Therefore, we have deliberately chosen to only include patients with oesophageal atresia type C. In our experience, about 90% of the patients with oesophageal atresia has type C (Vergouwe, Arch Dis Child, 2019, 104(2): 152-157; Pedersen Arch Dis Child, 2012, 97(3): 227-32), so most of the patients will be eligible for our

study. By only including patients with OA type C in this trial, the steroid group and control group will be as equal as possible.

However, this is a unique attempt to investigate the problem of postoperative strictures in a prospective study, as desperately suggested by multiple authors.

The protocol is detailed, the outcome parameters are accurate. Therefore, I don't have any major/minor concerns regarding the study design. My recommendation is to accept the manuscript.

Reviewer 2:

Thank you for giving me a chance to review this article. This is a study protocol of randomized controlled trial to evaluate whether intralesional steroid injections combined with endoscopic dilation can prevent refractory strictures in children with oesophageal atresia. I hope steroid injection would make a contribution to the reduction of dilations.

Major comments

A recent randomized controlled trial found that adding steroid injections before dilation did not result in a significant reduction of repeat dilations in patients with anastomotic strictures after esophagectomy (Herdes M et al. Endoscopic corticosteroid injections do not reduce dysphagia after endoscopic dilation therapy in patients with benign esophagogastric anastomotic strictures. Clin Gastroenterol Hepatol. 2013; 11:795-801.) However, another randomized controlled trial found that targeted steroid injection into the laceration after dilation reduce the need for repeat dilation (Hanaoka N et al. Endoscopic Balloon Dilation Followed By Intralesional Steroid Injection for Anastomotic Strictures After Esophagectomy: A Randomized Controlled Trial. Am J Gastroenterol. 2018;113:1468-1474.) However, no studies have compared injection before with injection after dilatation and this would be remained to be discussed. Why the authors use steroid injections prior to the dilatation?

We thank the second reviewer for evaluation of our manuscript and the suggestions for improvement. We agree that the timing of the injection in relation to the dilatation could remain a point of discussion. Unfortunately, intralesional steroid injections have been studied poorly up till now, especially in a randomized setting. The study the reviewer is referring to is in a different population with adult patients. We question to which extent this result can be extrapolated to paediatric patients with oesophageal atresia.

Therefore, we have based our decision to inject the steroids prior to dilation on the following. In children, no literature is available about targeted steroid injections into the laceration after dilatation. Stricture formation after oesophageal atresia repair often occurs in the first year of life, automatically leading to very young patients with a fragile oesophagus and a small lumen. Good visualization of the stricture when injecting the steroids is crucial to prevent perforation. We believe that this is best possible when injecting the steroids prior to dilatation. Since Kenacort is a slow-release medicine, we expect its effect to last after the dilatation.

Minor comments

1. According to the Patient timeline, the treating physician will decide on a third dilatation on the basis of the clinical signs of dysphagia and the findings on the esophagram. What are the clinical signs of

dysphagia and findings on the esophagram? Are there any criterions for third and additional dilations such as dysphagia score? The authors should clarify the indication of third and additional dilatations precisely.

Need details on your dysphagia scale and is it a validate score?

The Montreal Feeding Scale is the only internationally validated dysphagia scale available for children (Ramsay, *Paediatr Child Health*, 2011, 16(3): 147-e17; van Dijk, *Netherlands Journal of Psychology*, 2011, 66: 112-119). We will use this instrument to obtain dysphagia scores at the end of the follow up period. However, this instrument only has been validated from the age of six months old and is therefore not suitable for younger infants. Stricture formation after oesophageal atresia repair often occurs in the first year of life, and in our experience mostly in the first six months after surgical correction (Vergouwe, *Arch Dis Child*, 2019, 104(2): 152-157). Thus this dysphagia score is not suitable as a criterion for the 3rd dilation.

For that reason, we have chosen a combination of subjective (clinical signs) and objective (oesophagram) criteria to determine the indication for a 3rd dilatation. We added the following sentence on page 9, row 232-234: "Clinical signs of dysphagia are defined as the inability to be fed age-appropriately. Findings on the oesophagram indicating a stricture are defined as a significant narrowing of the lumen, seen as a waist in the contrast on the x-ray."

2. In method and analysis, the authors mentioned that the intervention is an endoscopic injection of Kenacort in each quadrant of the stricture prior to the third endoscopic dilatation. However, the order of injection and dilation was reversed in abstract. According to the abstract, the patients will be randomized into balloon dilatation followed by intralesional steroid injection or dilation only.

We apologize for this omission. In the STEPS-EA trial the intervention will indeed be an endoscopic injection prior to the 3rd dilatation. We changed the Abstract on page 2, row 44-46: "Children with oesophageal atresia type C (n=110) will be randomised into intralesional steroid injection followed by balloon dilatation or dilatation only."

3. Were the multiple triamcinolone injections allowed - or just the initial treatment? I suspect initial, but need to clarify.

In order to equally compare the study groups, no (additional) injections are allowed during the follow up period after the 3rd dilatation. Next to the effectiveness, this study also investigates the safety of the intervention. Kenacort is registered as a drug for children above the age of 12 years old. It has not been investigated in a randomized setting in children before. That's a second reason why, for now, we do not allow additional (multiple) injections.

To clarify this, we have added the following sentences on page 9, row 243-246: "During a follow up period of six months, patients will not receive any (additional) steroid injections, but only balloon dilatations if needed. Meaning, patients in the steroid group will only receive one injection, and patients in the control group will receive no injections at all. After the study period has ended, treatment is again free of choice." Also, we have added 'one-time' to the description of the investigational product on page 9, row 220.

VERSION 2 – REVIEW

REVIEWER	Noboru Hanaoka Osaka Red Cross Hospital, Osaka, Japan
REVIEW RETURNED	07-Nov-2019

GENERAL COMMENTS	Thank you for revising this manuscript. I hope the hypothesis of this trial will be proved.
---

VERSION 2 – AUTHOR RESPONSE

Reviewer(s)' Comments to Author:

Reviewer: 2

Reviewer Name

Noboru Hanaoka

Institution and Country

Osaka Red Cross Hospital, Osaka, Japan

Please state any competing interests or state 'None declared':

None

Please leave your comments for the authors below Thank you for revising this manuscript. I hope the hypothesis of this trial will be proved.

Thank you for your positive response to our revision.